# The Relationship between Bullshit Receptivity and Willingness to Share Misinformation about Climate Change: The Moderating Role of Pregnancy [note 1]

**DOI:** 10.3390/ijerph192416670

**Published:** 2022-12-12

**Authors:** Kaisheng Lai, Yingxin Yang, Yuxiang Na, Haixia Wang

**Affiliations:** School of Journalism and Communication, Jinan University, Guangzhou 510632, China

**Keywords:** pregnancy, bullshit receptivity, belief in misinformation about climate change, willingness to share misinformation about climate change

## Abstract

Widespread dissemination of misinformation about climate change has seriously harmed the health of future generations and the world. Moreover, misinformation-sharing behaviors exhibit strong individual characteristics. However, research is limited on the antecedents of and mechanism underlying the willingness to share misinformation about climate change in terms of individual personalities and physiological states. Accordingly, we surveyed 582 women (224 pregnant) using a questionnaire and constructed a moderated mediation model to explore the relationships among individuals’ bullshit receptivity, belief in misinformation about climate change, willingness to share misinformation about climate change, and pregnancy. The results showed that: (1) bullshit receptivity is positively related to the willingness to share misinformation about climate change; (2) belief in misinformation about climate change mediates the relationship between bullshit receptivity and willingness to share misinformation about climate change; and (3) for individuals with higher bullshit receptivity, pregnancy exacerbates the detrimental effects of bullshit receptivity on belief in misinformation about climate change.

## 1. Introduction

Although the scientific community has basically reached a scientific consensus on the fact that climate change is caused by human beings [1], there is still and will continue to be much misinformation about climate change, resulting in serious public health consequences [2,3]. For example, an investigative report by the nonprofit Stop Funding Heart showed that after analyzing a dataset of more than 195 Facebook pages and groups, approximately 45,000 posts downplayed or denied man-made climate change. The number of views ranged from 818,000 to 1.36 million, and the average number of interactions per post increased by 76.7% during 2021 [4]. The widespread misinformation about climate change will exacerbate political polarization [5], interfere with the public’s scientific consensus on climate change [6,7], and hinder the implementation of public policies [8]. Indeed, it is clear that misinformation about climate change hinders us from taking action to deal with the social dilemmas associated with climate change.

Less clear, however, are the personalities associated with misinformation about climate change as well as the physical causes and psychological mechanism underlying this misinformation. Exploring such antecedents and mechanisms of misinformation about climate change is important because such knowledge can be leveraged to increase the intention to address climate change for our future generations’ health. On the one hand, it is beneficial to theoretically understand the antecedents of and internal mechanisms underlying misinformation about climate change; on the other hand, it is also beneficial to take practical measures to intervene in antecedent variables and mediating mechanisms to reduce the spread of misinformation. Therefore, exploring misinformation about climate change has both theoretical and practical value.

In this study, we explored the personality and physiological antecedent variables of willingness to share misinformation about climate change from individual personality and physiological perspectives. First, previous research has shown that bullshit receptivity may be an important antecedent personality variable [9]. Over-acceptance of pseudoprofound bullshit is associated with increased belief in religious, paranormal, conspiratorial, and dubious health-related claims [10]. Second, the willingness to share misinformation about climate change is closely related to individual characteristics. The literature on the characteristics of populations that are susceptible to misinformation about climate change has focused on people with a more extreme and right-wing political orientation [11,12,13,14,15], college students [16,17], and older individuals [18,19]. However, as they are directly related to our children’s future health, the critical group of pregnant women has mostly been overlooked in the misinformation about climate change literature. Thus, little is known about the relationship between bullshit receptivity and the willingness to share misinformation about climate change in pregnant women.

Therefore, this study explored some pregnant women overly and non-skeptically accepting a wide variety of claims through the “reflexive open-mindedness” characteristic of pregnant women [10] (p. 19). We proposed the following questions and constructed a moderated mediation model (Figure 1): (1) What is the impact of bullshit receptivity on the willingness to share misinformation about climate change? (2) Does belief in misinformation about climate change mediate the relationship between bullshit receptivity and willingness to share misinformation about climate change? (3) Does pregnancy moderate the relationship between bullshit receptivity and belief in misinformation about climate change? In other words, is this mediating effect stronger among some pregnant women?

### 1.1. Definition of Misinformation

Before conducting our research, we need to clarify the definition of misinformation. Many researchers have distinguished between misinformation and disinformation [20,21]. Misinformation is generally considered to be incorrect or misleading, possibly as a result of human error. Disinformation is false and deceptive information disseminated with a clear intent to cause harm; for example, some interest groups that organize disinformation campaigns claim that climate change is not happening or that it is not caused by humans and deliberately conceal the threats posed by climate change to people and the environment [22,23]. According to previous research, it is difficult to predict the deceptive intention of the public to create or share misinformation in real life, and misinformation may be misreported by people who do not know the truth. Therefore, in our study, the following definition is used: “Misinformation is misleading information that is created and spread, regardless of whether there is intent to deceive” [2] (p. 3).

### 1.2. The Relationship between Bullshit Receptivity and Willingness to Share Misinformation about Climate Change

The concept of bullshit proposed by Frankfurt in 1986 [24] has been applied to various research fields, such as philosophy, economics, psychology, communication, and public policy. Bullshit claims are seemingly profound but actually meaningless and have no concern for the truth.

Furthermore, it extends to the field of misinformation, which means that an individual’s bullshit receptivity may have a direct impact on the sharing of misinformation [25]. When individuals are exposed to pseudo-profound bullshit, those who are not highly skeptical of dubious beliefs are more likely to fall prey to “conspiracy theories” (e.g., misinformation about climate change), resulting in the sharing of misinformation about climate change [10]. In contrast, individuals with low pseudo-profound bullshit receptivity may be more likely to have high skepticism and analytical thinking abilities that enable rational behavioral decisions, thereby decreasing the frequency and probability of misinformation-sharing behavior. Thus, we propose the following:

**H****1:** *Bullshit receptivity is positively related to the willingness to share misinformation about climate change*.

### 1.3. The Mediating Role of Belief in Misinformation about Climate Change

Previous studies have suggested that people with higher bullshit receptivity are more gullible, thereby increasing their belief in dubious claims [9]. In addition, bullshit receptivity is related to analytical reasoning ability [10]. Individuals with higher bullshit receptivity are likely to believe misinformation to be profound and thus believe misinformation. In contrast, individuals with low bullshit receptivity are more likely to have higher levels of skepticism and can better distinguish between true and false information through strict logical analysis.

We further argue that belief in misinformation about climate change is associated with the willingness to share misinformation about climate change. Previous studies have shown that people are not only more likely to accept ideas that are consistent with their own thoughts, perceptions, and beliefs [26], but also more inclined to share propositions that are consistent with their own preferences, attitudes, and positions with others [27,28]. Overall, when individuals believe a claim is reliable and correct, they may enhance their emotional identification with the claim, leading to sharing it with others. In contrast, when individuals perceive that information is misleading, they may reduce the sharing of misinformation. Thus, combining H1, we propose the following:

**H****2:** *Belief in misinformation about climate change mediates the relationship between bullshit receptivity and willingness to share misinformation about climate change*.

### 1.4. The Moderating Role of Pregnancy

Pregnancy is an extremely challenging event for women [29] and may affect their perceptions of the world [30]. Women begin to gradually complete the transition to motherhood [31] and increase their focus on their own health and the health of the next generation [32].

Some scientific studies have shown that the short-term memory and verbal memory of pregnant women may be slightly affected, resulting in difficulty concentrating, especially in the third trimester. In addition, their own health, the health of their unborn child, and their relationship with their partners may affect the sense of uncertainty of pregnant women [33,34] and enhance their sensitivity to various threats and injuries in the environment [35], which may lead to high levels of stress, anxiety, and fear among pregnant women [36]. These negative emotions and low attention levels may interfere with some pregnant women’s rational decision-making, which may lead them to overbelieve some claims unintentionally and further influence their belief in misinformation. Accordingly, for individuals with higher bullshit receptivity, pregnancy exacerbates the detrimental effects of bullshit receptivity on belief in misinformation about climate change. Thus, we propose the following:

**H****3:** *Pregnancy moderates the relationship between bullshit receptivity and belief in misinformation about climate change such that this relationship is stronger for some pregnant women*.

## 2. Materials and Methods

### 2.1. Sample and Procedure

Due to the limitations of epidemic prevention and control measures, 628 female participants were recruited for this study using an online questionnaire collection platform. To ensure that the participants answered carefully, we conducted an attention test with a screening question: “Are you female?” Based on this question, 46 invalid samples were removed, and 582 valid samples were finally obtained, including 224 pregnant women and 358 nonpregnant women (including those who had never given birth and those who had given birth but were not currently pregnant). The sample varied in demographic characteristics; for example, 224 women (38.5%) were in the pregnant group with a mean age of 29.85 (SD = 3.78), and 358 women (61.5%) were in the nonpregnant group with a mean age of 29.60 (SD = 7.26) (Table 1). The online questionnaire system required each participant to complete all questions before submitting the questionnaire. The IP address of each participant was recorded in the background, and each IP had only one submission opportunity. Therefore, there was no partial loss of data or repeated responses from the same participant.

The survey was conducted after the participants completed the informed consent form. In this survey, the participants were asked to indicate their belief in and willingness to share misinformation about climate change. Then, they completed the bullshit receptivity scale. Finally, the participants answered additional demographic questions that had been added to the questionnaire. After completing the survey, the participants were told that “The claim about climate change in the above questionnaire has been identified as misinformation by fact-checking websites.” Personal information was not collected, analyzed, or presented in this study.

### 2.2. Measures

All scales used in this study are international maturity scales used in previous studies. We follow the translation/back-translation procedure to create the measures in Chinese [37]. Unless otherwise specified, all participants responded on a 5-point Likert scale. Based on previous similar studies [38,39,40,41], we treated the 5-point scale with parametric tests.

#### 2.2.1. The Belief and Willingness to Share Misinformation about Climate Change

Based upon our definition of misinformation, we selected five statements of misinformation about climate change identified by fact-checking websites. Three were taken from FactCheck.org, a nonprofit and nonpartisan fact-checking website in the United States; the other two were taken from Tencent Jiaozhen, a professional and timely fact-checking platform in China. Given that the majority of the public reads only article headlines when exposed to information on social media [42], we followed previous research on misinformation by presenting only the headlines of the misinformation rather than the full articles [9,43]. For example, “Scientific research shows that global warming is a conspiracy theory, and global warming has stopped in the past few decades.” After reading each headline, the participants were asked the following questions (in the following order): “To the best of your knowledge, how accurate is the claim in the above headline?” (1 = not at all accurate, 5 = extremely accurate); and “Would you consider sharing this story online (for example, through WeChat or Micoe-blog)” (1 = not at all consider, 5 = extremely consider) [9].

#### 2.2.2. Bullshit Receptivity

Following previous research, we measured bullshit receptivity using a 10-item bullshit receptivity (BSR) scale [10]. The participants were shown ten randomly generated sentences that were filled with abstract buzzwords and randomly constructed according to a certain syntactic structure. These statements seemed profound but were actually constructed without concern for the truth. The participants were asked to read each statement and rate how “profound” the statement was. “Profound” means “of deep meaning; of great and broadly inclusive significance” (1 = not at all profound, 5 = extremely profound). The bullshit receptivity score was the mean of the profoundness ratings for all bullshit items. A sample item was as follows: “Hidden meaning transforms unparalleled abstract beauty”. The internal consistency of the instrument was measured by Cronbach’s α, which was 0.654. According to previous research, an alpha above 0.6 is acceptable [44,45,46].

#### 2.2.3. Control Variables

We measured the participants’ age, education level, employment status, marital status, and number of children as control variables because previous studies have confirmed that age and education level are usually related to individuals’ cognitive level and media literacy. Employment status, marital status, and the number of children may affect an individual’s time and energy levels. To prevent the interference of these variables, this study used them as control variables.

#### 2.2.4. Data Analysis

In this study, we used SPSS 25 software (IBM Corp., Armonk, NY, USA) to perform Harman’s single-factor test and Hayes’ PROCESS software to examine our proposed moderated mediation model [47]. In PROCESS, the ordinary least regression function enables the statistical testing of mediation, moderation, and moderated mediation models. PROCESS has frequently been used in the fields of psychology, business, communication, and health sciences for hypothesis testing. Based on our proposed model, model 4 was used to test the simple mediation model. If the bias-corrected bootstrap 95% confidence interval (CI) did not include zero, it indicated a significant mediation effect at the level of α= 0.05 [47]. This analysis allowed us not only to test the direct and indirect relationship between bullshit receptivity and willingness to share misinformation about climate change, but also to spot whether belief in misinformation about climate change has mediating roles in this relationship. Model 7 was selected to test our first-stage moderated mediation model. Specifically, we determined whether the variable of pregnancy played a moderating role in the mediating path. Bootstrapping with 5000 resamples was employed to test the significance of our proposed hypothesis.

## 3. Results

### 3.1. Preliminary Analysis

Before testing our hypothesis, we conducted a preliminary analysis. Harman’s single factor test was used to determine the variance for the single-factor solution (variance = 19.89%, i.e., <40%), which indicated that the present research was not affected by common method variance (CMV) [48]. The descriptive statistics and correlation matrix pertaining to our focal variables are shown in Table 2. The willingness to share misinformation about climate change was positively correlated with bullshit receptivity, belief in misinformation about climate change, and pregnancy (Table 2). Subsequently, we conducted a regression analysis to test Hypothesis 1 (Table 3). In support of Hypothesis 1, bullshit receptivity was positively related to the willingness to share misinformation about climate change (*b*_1_ = 0.382, *p* < 0.001); this effect persisted after controlling for age, education, employment status, marital status, and the number of children born (*b*_2_ = 0.381, *p* < 0.001). Moreover, education (*b*_3_ = −0.134, *p* < 0.05) was a significant predictor of the dependent variable (willingness to share misinformation) but age, employment status, marital status, and the number of children born were not statistically significant predictors.

### 3.2. Model Testing

Hypothesis 2 posited that bullshit receptivity has a positive indirect effect on the willingness to share misinformation about climate change. We used model 4 of the PROCESS macro in SPSS for this analysis. When the mediator variable (belief in misinformation about climate change) was added, the direct effect of bullshit receptivity on willingness to share misinformation about climate change was significant (*b*_4_ = 0.158, *p* = 0.0042); bullshit receptivity had a significant predictive effect on belief in misinformation about climate change (*b*_5_ = 0.313, *p* < 0.001); and belief in misinformation about climate change had a significant predictive effect on willingness to share (*b*_6_ = 0.718, *p* < 0.001). The mediation analysis results for the effect of bullshit receptivity on the willingness to share misinformation about climate change via belief in misinformation about climate change revealed that this indirect effect was significant (*b*_7_ = 0.225, 95% confidence interval (CI) = [0.132, 0.324]) when using the bias-corrected bootstrap CIs. As shown in Table 4 and Table 5, after controlling for the participants’ age, education, employment status, marital status, and the number of children born, the direct effect was still significant (*b*_8_ = 0.148, *p* = 0.009). The predictive effect of bullshit receptivity on belief in misinformation about climate change (*b*_9_ = 0.326, *p* < 0.001) and belief in misinformation about climate change on willingness to share misinformation about climate change (*b*_10_ = 0.714, *p* < 0.001) were still significant; the indirect effect was still significant (*b*_11_ = 0.233, 95% CI = [0.144, 0.333]). The mediating effect was significant and partial. The mediating effect accounted for 61.15% of the total effect. Thus, Hypothesis 2 was supported.

Hypothesis 3 posited that pregnancy serves as a first-stage moderator of the mediation effect of bullshit receptivity on willingness to share misinformation about climate change via belief in misinformation about climate change. As expected, both the mediator variable model (*F* (8, 573) = 5.705, R^2^ = 0.074, *p* < 0.001) and the dependent variable model (*F* (7, 574) = 56.618, R^2^ = 0.408, *p* < 0.001) were significant after controlling for age, education, employment status, marital status, and the number of children. As shown in Table 6, bullshit receptivity and pregnancy predicted belief in misinformation about climate change (*b*_12_ = 0.255, *p* < 0.05). Figure 2 depicts the relevant interaction plot. As shown in Figure 2, although bullshit receptivity was positively related to belief in misinformation about climate change for the nonpregnant participants (*b*_13_ = 0.230, *p* < 0.05), this effect was lower than the conditional effect for the pregnant participants (*b*_14_ = 0.485, *p* < 0.001).

In addition to the interaction, the results further supported a significant moderated mediation model according to which the association between bullshit receptivity and willingness to share misinformation about climate change (as mediated by belief in misinformation about climate change) was further moderated by pregnancy. For the pregnant participants, the indirect effect was significant, and the effect was stronger for these users (*b*_15_ = 0.34, 95% CI = [0.204, 0.497]) than for the nonpregnant participants (*b*_16_ = 0.16, 95% CI = [0.057, 0.281]). These results showed that pregnancy strengthens the positive association between bullshit receptivity and belief in misinformation as well as the mediating effect of belief in misinformation on the relationship between bullshit receptivity and willingness to share misinformation about climate change. Thus, Hypothesis 3 was supported. Moreover, age (*b*_17_ = 0.011, *p* < 0.05) was a significant predictor in the mediator variable (belief in misinformation) model but education, employment status, marital status, and the number of children born were not statistically significant predictors. Employment status (*b*_18_ = −0.184, *p* <0.05) was a significant predictor in the dependent variable (willingness to share misinformation) model but age, education, marital status, and the number of children born were not statistically significant predictors.

## 4. Discussion

With the popularity of social media, the reach, growth rate, and potential harm of misinformation have increased [49]. The widespread dissemination of misinformation about climate change is not only detrimental to building a scientific consensus on climate change, but also a serious threat to future public health. Therefore, it is currently an important social issue to fully understand the kind of people who are more susceptible to misinformation about climate change to effectively manage misinformation about climate change [50].

Our first main finding was that people who are more receptive to bullshit are more likely to believe misinformation about climate change to be accurate and more likely to share misinformation about climate change. This result is consistent with previous theoretical and empirical findings [9]. Bullshit receptivity is often associated with cognitive ability. According to dual-process theory, there are two ways for people to process information. One is analytic thinking, which needs to mobilize the individual’s complex cognitive ability to conduct deep and deliberative information processing to make more rational decisions. The other is intuitive thinking, which mainly relies on the individual’s rapid and autonomous responses [14,51,52]. Therefore, people who are more receptive to bullshit rely more on intuitive thinking and shallow processing of information. They lack skepticism, leaving them more likely to be deceived by misinformation. In contrast, people who are less receptive to bullshit rely more on analytical thinking and have a better discrimination ability for pseudo-profound bullshit, which seems to be true and profound but is actually meaningless. In addition, people who are more receptive to bullshit are more likely to believe in conspiracy theories [10,53]. In contrast, people who are less receptive to bullshit are more skeptical of religion and paranormal phenomena [54] and more inclined to accept evolutionism than creationism [55].

Second, an interesting finding was that bullshit receptivity can directly influence the willingness to share misinformation about climate change. As previous studies have shown, people sometimes do not fully consider the accuracy of information in the decision-making process of sharing information on social media [28], which may explain why misinformation generally spreads faster and wider and is accepted more easily than accurate information [56]. According to social motivation theory, giving individuals legitimate motivations and channels prompts them to share behaviors on social media [57]. However, there are many factors driving people’s information-sharing behaviors, including the maintenance of personal interests or reputation [58], social interaction [59], and interest in information [60]. According to social communication theory, individuals’ behavioral decisions in social relations are based on the maximization of individual interests. In addition, research has proven that people often share information that “is interesting-if-true”, even though the information itself may be incorrect [60].

Third, we found that pregnancy strengthens the positive relationship between bullshit receptivity and belief in misinformation about climate change and that belief in misinformation mediates the relationship between bullshit receptivity and willingness to share misinformation about climate change. A previous study showed that skeptical and analytical thinkers are better at distinguishing between true and false information [14]. In contrast, people who overly and un-skeptically accept a wide variety of claims tend to be more gullible [10], a trait known as reflexive open-mindedness [9]. This study found that the “reflexive open-mindedness” trait is more significant for some pregnant women. Moreover, if someone was older or had a busy work schedule, they were more likely to misread this information without filtering, which was consistent with previous research [38].

In addition, it is necessary to make it clear that the purpose of our research is not to stigmatize pregnant women. In contrast, we aim to better understand women by revealing the psychological, physiological, and behavioral changes brought about by this special physiological period of pregnancy. First, pregnant women are faced with a series of changes in stress, hormones, and brain structure and function, which have many complex influences on pregnant women’s physiology, psychology, and behavior, including negative and positive effects [31]. On the one hand, some scholars have discussed the positive effects of pregnancy. For example, with regard to decision-making about future generations, pregnant women prefer long-term decision-making compared with nonpregnant women [61]. On the other hand, it is equally important to discuss the potential negative effects that women may face in this special physiological period of pregnancy. Second, the concept of bullshit originated from philosophy [24], and some evidence from psychology had shown that it was linked with the willingness to share misinformation [9,14]. Based on this theoretical thinking and empirical evidence, our study focused on the female perspective for further discussion of the consequences of bullshit receptivity. Third, our research results represented only a correlational rather than a causal conclusion for some individuals (the number of women whose BSR score was greater than the average plus one standard deviation was relatively small, accounting for 18%). Our results implied that a small number of pregnant individuals may interact with bullshit receptivity, resulting in some negative consequences. Particularly, for individuals with higher bullshit receptivity, pregnancy exacerbated the detrimental effects of bullshit receptivity on belief in misinformation about climate change. Therefore, these pregnant women deserve our special attention as their situation has been neglected in the past research. Thus, our research is important and necessary.

Our research has practical implications for the governance of misinformation about climate change. The premise of governance is to identify the populations susceptible to misinformation about climate change. Only a full understanding of the characteristics of susceptible populations enables the effective prevention of the dissemination of misinformation. We need timely and effective help for individuals who are more receptive to bullshit by increasing their healthy skepticism and strengthening scientific consensus on climate change, thereby reducing the negative impact of misinformation on them.

Admittedly, there are some limitations to this study that are worth considering. First, future research can further expand the analysis of the effect and internal mechanism of pregnancy on the willingness to share misinformation about climate change; for example, exploring the susceptibility of women at various stages of pregnancy to misinformation about climate change. Second, the state and characteristics before and after pregnancy, the situation when the information is specially toned down, or the seriousness of the information may all affect the robustness of the results, which should be given special attention in future research. Third, this study took place in China. Given the cultural differences across countries, we believe that considering the role of culture when investigating the psychological motivations underlying the willingness to share misinformation about climate change could be a fruitful direction for future research. In addition, we need to pay attention to the fact that culture may affect the willingness to share misinformation about climate change through the perception of the phenomenon of climate change itself.

In addition, it should be noted in our original design that we also expected the association between pregnancy and willingness to share fake news about climate change to be mediated by the susceptibility to future generations’ health. Additionally, among those with lower levels of media truth discernment, the positive indirect effect of pregnancy on the willingness to share fake news on climate change is stronger. However, these results were not supported by the results, and the study found that the effect of pregnancy on the susceptibility to future generations’ health was not significant, which may be due to the limitations of the sensitivity of the questionnaire itself. Therefore, a more sensitive measurement method can be explored in future research, such as employing an experimental method.

Regarding ethics and data privacy, with the convenience and sharing of online questionnaire collection platforms, there were many mature and secure online questionnaire-collection platforms that have greatly improved the reach and popularity of questionnaires and maximized the scientific validity of sampling in terms of participant selection. Therefore, due to the restrictions caused by epidemic prevention, we used an online questionnaire-collection platform. Since the current relevant industry laws and regulations in China do not explicitly require ethical review for this type of research, all of the scales used in our study were mature scales that have been used in the past. Therefore, this study did not require approval from the Institutional Review Board.

We attached great importance to the personal privacy of our participants and respected moral and ethical norms. According to the basic privacy protection regulations, our study adopted appropriate protection methods to protect the participants’ private data. First, informed consent was obtained from the participants for all our research. Second, this study used anonymous data collection. Only information relevant to this study, such as the participants’ beliefs, attitudes, and willingness to share behaviors related to climate change information and basic demographic variables was collected. Other private personal information was not collected. Third, at the level of behavioral decision-making, we collected only the participants’ willingness to share climate change information and did not investigate actual sharing behaviors in the past. Fourth, no ethical comments were made on the results of the participants’ scores. Fifth, we used the collected information only for academic research and kept it strictly confidential. Sixth, at the end of the questionnaire, the participants were informed that the headlines had been falsified by an official fact-checking platform to prevent the participants’ misunderstanding as an ethical consideration.

The continuous improvement of laws related to privacy protection and data security in China will change the way information is acquired on online data-collection platforms and enable better protection of participants’ privacy. In the future, we will further strengthen the review of ethics and morality.

## 5. Conclusions

In this study, we focused on the female perspective by exploring some pregnant women overly and un-skeptically accepting a wide variety of claims about climate change. Therefore, we investigated the relationships among bullshit receptivity, belief in misinformation about climate change, willingness to share misinformation about climate change, and pregnancy. The results showed that people with higher bullshit receptivity are more likely to believe in misinformation about climate change, resulting in sharing behavior about climate change. Belief in misinformation partially mediated the relationship between bullshit receptivity and willingness to share misinformation about climate change. Moreover, for individuals with higher bullshit receptivity, pregnancy exacerbated the detrimental effects of bullshit receptivity on belief in misinformation about climate change.

## Figures and Tables

**Figure 1 ijerph-19-16670-f001:**
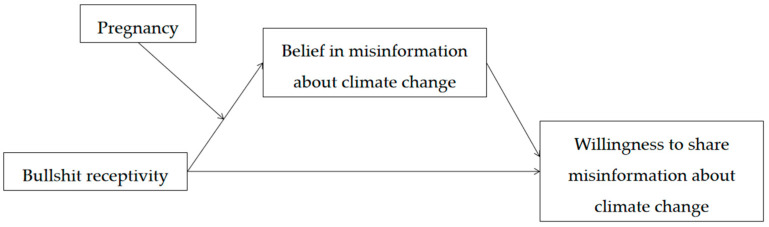
The hypothesized moderated mediation model.

**Figure 2 ijerph-19-16670-f002:**
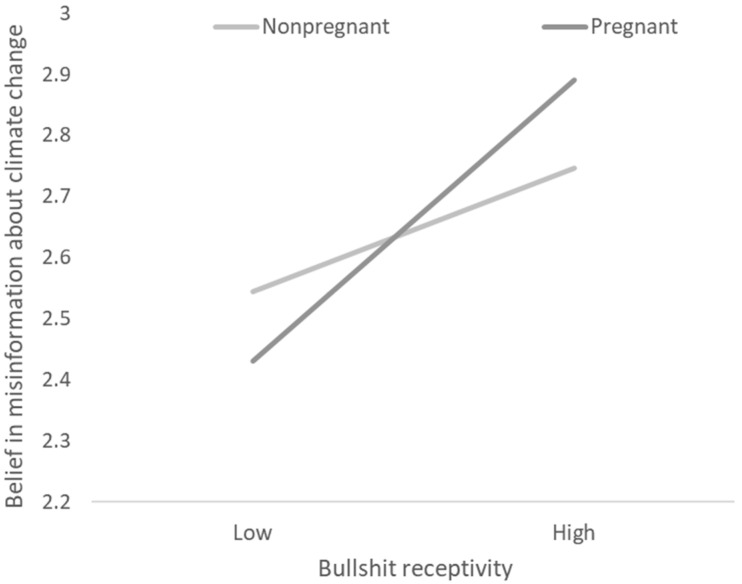
The effect of the two-way interaction between bullshit receptivity and pregnancy on belief in misinformation about climate change.

**Table 1 ijerph-19-16670-t001:** Sample characteristics (*n* = 582).

Demographic Variables		Pregnant Group	Nonpregnant Group
		*n* (%)	*n* (%)
Age	Mean (S)	Mean = 29.85 (SD = 3.78) 22–42	Mean = 29.60 (SD = 7.26) 13–65
Employment status	Employed	204 (91.1%)	298 (83.2%)
	Unemployed	20 (8.9%)	60 (16.8%)
Education	Less than technical secondary school	3 (1.3%)	13 (3.6%)
	Some college	18 (8.0%)	40 (11.2%)
	Bachelor’s degree	180 (80.4%)	279 (77.9%)
	Master’s degree	21 (9.4%)	24 (6.7%)
	Doctor’s degree or higher	2 (0.9%)	2 (0.6%)
Marital Status	Married	221 (98.7%)	237 (66.2%)
	Unmarried	3 (1.3%)	119 (33.2%)
	Other	0	2 (0.6%)
The number of children born	0	71 (31.7%)	132 (36.9%)
	1	129 (57.6%)	172 (48.0%)
	2	23 (10.3%)	51 (14.2%)
	3	1 (0.4%)	3 (0.8%)

**Table 2 ijerph-19-16670-t002:** Descriptive statistics and intercorrelations among variables (*n* = 582).

Title 1	M	SD	1	2	3
Bullshit receptivity	3.689	0.479	-		
Belief in misinformation	2.658	0.669	0.224 **	-	
Willingness to share misinformation	2.202	0.795	0.230 **	0.625 **	-
Pregnancy	0.380	0.487	0.108 **	0.040	0.092 *

Note. Pregnancy was coded as follows: 0 = Nonpregnant, 1 = Pregnant. * *p* < 0.05, ** *p* < 0.01.

**Table 3 ijerph-19-16670-t003:** Regression analysis of dependent variable model (*n* = 582).

	b	SE	t	*p*
**Model 1**				
Constant	0.791 **	0.250	3.169	<0.01
Bullshit receptivity	0.382 ***	0.067	5.698	<0.001
**Model 2**				
Constant	1.074 *	0.491	2.188	<0.05
Age	0.008	0.006	1.260	0.208
Education	−0.134 *	0.060	−2.248	<0.05
Employment status	−0.126	0.106	−1.186	0.236
Marital status	0.010	0.106	0.096	0.924
The number of children born	0.006	0.063	0.090	0.929
Bullshit receptivity	0.381 ***	0.068	5.561	<0.001

Note. Unstandardized regression coefficients are reported. Bootstrapping sample size = 5000. * *p* ≤ 0.05, ** *p* ≤ 0.01, *** *p* ≤ 0.001.

**Table 4 ijerph-19-16670-t004:** Regression analysis of the mediation model (*n* = 582).

	b	SE	t	*p*
**Mediator variable (Belief in misinformation) model**				
Constant	1.182 **	0.414	2.852	<0.01
Age	0.011 *	0.005	2.069	<0.05
Education	−0.069	0.050	−1.371	0.171
Employment status	0.082	0.089	0.917	0.360
Marital status	0.033	0.089	0.372	0.710
The number of children born	0.007	0.053	0.138	0.890
Bullshit receptivity	0.326 ***	0.058	5.649	<0.001
**Dependent variable (Willingness to share misinformation) model**				
Constant	0.231	0.395	0.584	0.560
Age	0.0001	0.005	0.017	0.986
Education	−0.085	0.048	−1.778	0.076
Employment status	−0.184 *	0.085	−2.175	<0.05
Marital status	−0.014	0.084	−0.160	0.873
The number of children born	0.0004	0.050	0.008	0.994
Bullshit receptivity	0.148 **	0.056	2.630	<0.01
Belief in misinformation	0.714 ***	0.039	18.084	<0.001

Note. Unstandardized regression coefficients are reported. Bootstrapping sample size = 5000. * *p* ≤ 0.05, ** *p* ≤ 0.01, *** *p* ≤ 0.001.

**Table 5 ijerph-19-16670-t005:** Bootstrapping analysis of the mediation model (*n* = 582).

	Effect	SE	BootLLCI	BootULCI
Direct effect	0.148	0.056	0.037	0.258
Indirect effect	0.233	0.048	0.144	0.333
Total effect	0.381	0.068	0.246	0.515

**Table 6 ijerph-19-16670-t006:** Conditional process analysis (*n* = 582).

	b	SE	t	*p*
**Mediator variable (Belief in misinformation) model**				
Constant	2.304 ***	0.338	6.816	<0.001
Age	0.011 *	0.005	2.124	<0.05
Education	−0.065	0.050	−1.289	0.198
Employment status	0.717	0.089	0.802	0.422
Marital status	0.058	0.100	0.576	0.564
The number of children born	0.018	0.054	0.333	0.739
Bullshit receptivity	0.230 **	0.073	3.170	<0.01
Pregnancy	0.043	0.633	0.676	0.50
Bullshit receptivity x Pregnancy	0.255 *	0.118	2.162	<0.05
**Dependent variable (Willingness to share misinformation) model**				
Constant	0.776 *	0.319	2.434	<0.05
Age	0.0001	0.005	0.017	0.986
Education	−0.085	0.048	−1.778	0.076
Employment status	−0.184 *	0.085	−2.175	<0.05
Marital status	−0.014	0.084	−0.160	0.873
The number of children born	0.0004	0.050	0.008	0.994
Bullshit receptivity	0.148 **	0.056	2.630	<0.01
Belief in misinformation	0.714 ***	0.039	18.084	<0.001
**Conditional effects of predictor (Bullshit receptivity) considering the moderator (Pregnancy)**	**b**	**BootSE**	**BootLLCI**	**BootULCI**
Nonpregnant	0.230 **	0.073	0.088	0.373
Pregnant	0.485 ***	0.094	0.302	0.669

Note. Unstandardized regression coefficients are reported. Bootstrapping sample size = 5000. * *p* ≤ 0.05, ** *p* ≤ 0.01, *** *p* ≤ 0.001.

## Data Availability

The data presented in this study are available on request from the corresponding author.

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
