# Peer review of "The Relationship between Bullshit Receptivity and Willingness to Share Misinformation about Climate Change: The Moderating Role of Pregnancy"

_ijerph, 2022, doi:10.3390/ijerph192416670_

Round 1

Reviewer 1 Report

The article deals with an essential topic of the possibility of duplicating unverified information about climate change by a selected social group, which are pregnant women.

My comments are as follows:

1)     Although the authors refer to the Frankfurt theory from 1986, I believe that the term bullshit for misinformation or nonsense is too strong and aggressive, especially for pregnant women who are under stress and hormonal changes.

2)     Conclusion number (3) given in the summary is, in my opinion, too far-reaching conclusion? Was being pregnant in the surveyed women associated with the fact that, compared to the period before pregnancy, these women were more inclined to nonsense? What about the situation when the information is specially toned down, or the seriousness of the information is reduced in the message to pregnant women in order not to upset them? Or on the contrary. Taking advantage of their susceptibility to worry and accepting misinformation, this can be used to gain negative benefits.

3)     In line 69 (as well as 398) the authors used the phrase "the dark side of pregnant women". It is inappropriate to use such a term. Pregnant women are not characters from science-fiction movies, nor are they unethical individuals. However, their behaviour can be manipulated.

4)     In section 1.4, in the sentence "For pregnant..." (lines 141-143), the authors incorrectly generalize the statement to all women.

5)     In section 2.2.2 authors should explain what α= 0.654 means. In a scientific article, values cannot be given without their interpretation.

6)     The authors should necessarily explain what model 4 and model 7 are. Understanding the presented solutions will then be easier for a wider group of readers.

7)     The authors made a significant analytical mistake. The measurement made on a 5-point Likert scale is an ordinal scale, where the median can be calculated, but the average cannot be counted (as required for Stevens scales). Only the 7-point scale is considered quasi-continuous. I understand that the authors have applied the solution schemes used by other authors to compare results.

8)     In sections 3.1 and 3.2, coefficient b repeatedly appears with different values. Since this coefficient denotes a parameter for a specific independent variable, it is necessary to add an appropriate, differentiating subscript each time.

9)     In sections 3.1 and 3.2, in the description of the implementation of research related to hypotheses 1 and 2, the values of the parameters from the estimated model appear. It is necessary to enter these values in the same way as in Table 3

10)  The authors should standardize the notation of all values. E.g. is (b =0.2330, 95% CI = [.1401, .3312]). Sometimes the integer value is entered, and sometimes it is missing. It should be (b = 0.2330, 95% CI = [0.1401, 0.3312]).

11)  Only some of the results from table 3 are discussed. So why were the other variables included in the model if their impact on the dependent variable is not discussed?

12)  In line 330, the authors write about "the previous study". What is the study about?

13) The "cultural differences" described in line 355 are very important. They also affect the perception of the phenomenon of climate change itself, which the authors should pay attention to.

Reviewer 2 Report

I was initially skeptical of this piece because it suggests a bias against pregnant women.  However, you controlled for many factors, which increased the study's validity.  The results are compelling.  I wonder if the duration of the pregnancy has any impact.

Reviewer 3 Report

The topic and the problematic field is very relevant and actual as well as very sensitive and emotionally highlighted. The article (structure, content and ect.) is good quality and give new understanding of the problem, as well as focus our attention to the prevention relevance. The conclusion that bullshit receptivity is positively related to the willingness to share misinformation about climate change, that belief in misinformation about climate change mediates the relationship between bullshit receptivity and willingness to share about climate change, and that pregnancy moderates the relationship between bullshit receptivity and belief in misinformation about climate change, with this relationship being stronger for pregnant women, are very important. The only concern is not to stigmatize pregnant women. 

Round 2

Reviewer 1 Report

I thank the authors for considering my remarks. I am satisfied with the changes made.